

# A retrospective study: does upper airway morphology differ between non-positional and positional obstructive sleep apnea?

Xiao Jiao, Jianyin Zou, Suru Liu, Jian Guan, Hongliang Yi and Shankai Yin

Otolaryngology, Shanghai Jiaotong University Affiliated Sixth People's Hospital, Shanghai, China
Otolaryngology Institute of Shanghai Jiao Tong University, Shanghai, China

## ABSTRACT

**Objective.** The objective of this study was to explore the differences in upper airway morphology between positional (POSA) and non-positional (NPOSA) obstructive sleep apnea.

**Methods.** This retrospective study enrolled 75 patients (45 NPOSA and 30 POSA) who underwent polysomnography (PSG) and computed tomography (CT). The differences in, and relationships of, the PSG values and CT data between POSA and NPOSA were analyzed.

**Results.** Significant ($p < 0.05$) differences between the two groups were found in the apnea/hypopnea index (AHI), lateral-AHI (L-AHI), soft palate length (SPL), cross-sectional palatopharyngeal area, and the coronal diameter (CD) of the palatopharyngeal area at the narrowest part of the glossopharynx, which were all higher in POSA, except for SPL, AHI, and L-AHI. L-AHI was correlated with the cross-sectional area ($r = -0.306$, $p = 0.008$) and CD ($r = -0.398$, $p < 0.001$) of the palatopharyngeal area, the cross-sectional area ($r = -0.241$, $p = 0.038$) and CD ($r = -0.297$, $p = 0.010$) of the narrowest level of the glossopharynx, the CD of the glossopharynx ($r = 0.284$, $p = 0.013$), body mass index (BMI, $r = 0.273$, $p = 0.018$), SPL ($r = 0.284$, $p = 0.014$), and vallecula-tip of tongue ($r = 0.250$, $p = 0.030$). The SPL and CD at the narrowest part of the glossopharynx were included in the simplified screening model.

**Conclusions.** In NPOSA, the CD of the upper airway was smaller, and the soft palate was longer, than in POSA. These differences may play significant roles in explaining the main differences between NPOSA and POSA.

Corresponding author
Hongliang Yi, yihongl@126.com

## INTRODUCTION

Obstructive sleep apnea-hypopnea syndrome (OSAHS) is harmful to health. Its major clinical features include snoring, apnea, and daytime hypersomnolence; it is also correlated with diabetes, ischemic heart disease, chronic cor pulmonale, and cerebrovascular disease. Based on differences in the apnea/hypopnea index (AHI) in different positions, OSAHS is divided into positional obstructive sleep apnea (POSA) and non-positional obstructive

apnea (NPOSA). POSA is defined as having an AHI in the supine sleeping position that is at least twice the value in other positions (*Oksenberg et al., 2012*); otherwise, the condition is considered to be NPOSA. NPOSA patients are prone to having more severe OSAHS than POSA patients (*Oksenberg et al., 1997*; *Oksenberg et al., 2012*). Additionally, a variety of treatment methods are available for OSAHS, but the effects of the these treatments on NPOSA and POSA vary (*De Vries et al., 2015*; *Lee et al., 2012*; *Levendowski et al., 2014*). However, the exact mechanisms underlying NPOSA and POSA are unclear; anthropometric characteristics, upper airway morphology, and anatomical structures may all play roles. Many studies have focused on the differences between NPOSA and POSA, such as age and body mass index (BMI) (*Oksenberg et al., 1997*; *Oksenberg et al., 2012*). There are differences in the contributions made by age and BMI to POSA (*Oksenberg et al., 1997*; *Teerapraipruk et al., 2012*). Only a few studies have compared the palatopharyngeal morphology (*Soga et al., 2009*), craniofacial structures and soft tissues of the lateral pharyngeal wall (*Saigusa et al., 2009*) between POSA and NPOSA in Caucasians. Differences in the upper airway morphology at other levels, and differences in the soft tissues of the upper airway (such as the length and thickness of the soft palate) between NPOSA and POSA are not known. Currently, the impact of these factors on the substantial differences evident between NPOSA and POSA is uncertain; there are especially few studies on Chinese populations. Therefore, to enhance our understanding of the pathogenesis of NPOSA and POSA, this study explored differences in the upper airway morphology and anatomical structures between two groups of Chinese patients using computed tomography (CT) and polysomnography (PSG).

## MATERIALS & METHODS

This project was approved by the Ethics Committee of our hospital and complied with all relevant tenets of the Declaration of Helsinki (2016-31-(1)). All participants provided written informed consent before inclusion in the study.

### Subjects

The study initially recruited 105 consecutive adult patients with OSAHS who underwent surgery because they refused CPAP in our hospital from January 2005 to December 2014 (BMI < 35 kg/m$^2$). Patients with chronic airway diseases, obstructive pulmonary disease, and systemic diseases were excluded. Eleven subjects were excluded because they refused to undergo CT testing or because the orbitale-porion plane (i.e., the Frankfort horizontal plane) was not parallel to the ground plane on CT of the head (*Yu, Fujimoto & Urushibata, 2003*). All patients were asked to complete the Epworth Sleepiness Scale and a sleep questionnaire, and to undergo overnight PSG. The study excluded 17 patients who slept for less than 30 min in the lateral or supine position (*Oksenberg et al., 1997*), and 2 patients whose central or mixed sleep-disordered breathing events accounted for more than 25% of all sleep-breathing episodes (*Strollo et al., 2014*). Ultimately, the study included 75 patients (Fig. 1).

### Methods

The head and neck multislice CT, overnight PSG test, and physical parameters were retrospectively reviewed in all subjects.

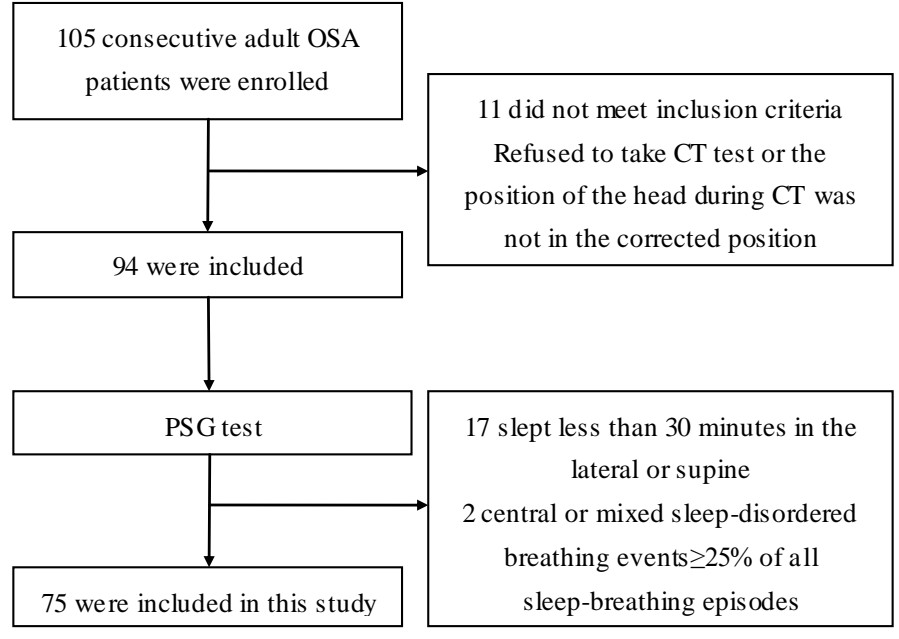

**Figure 1 Flow diagram of the recruitment of study participants.** PSG test, polysomnography test; CT, computed tomography.

### Body parameters

BMI was calculated by dividing the patient's weight in kilograms by height squared in meters squared. The tongue position and tonsil size were assessed using the Friedman staging system before overnight PSG.

### PSG test

All subjects underwent overnight PSG (Alice 4; Respironics, Pittsburgh, PA, USA), which not only identified sleep-disordered breathing events but also included a position sensor that could differentiate the sleep position as supine (S), left side (L), right side, and prone. The AHI is classified into three types depending on the sleep position: the overall AHI, AHI in the supine position (S-AHI), and AHI in the lateral position (L-AHI). The monitoring time was >7 h. The records were analyzed by professionals the next morning. Patients with OSHA who had AHI values in the supine sleeping position at least two-fold greater than the values in other positions were defined as having POSA; the other OSHA patients were defined as having NPOSA (*Oksenberg et al., 2012*). No subjects consumed strong tea, coffee, or drugs that would have had calming effects or caused sleep before the PSG test.

### CT analysis

CT (peak voltage, 120 kV; tube current, 130 mA; pitch, 0.75; section thickness, 2 mm; and tube current-time product, 100 mA) was performed on all patients in the supine position, from the skull base to the hyoid bone, with the head positioned correctly (the orbitale-porion plane parallel to the ground plane), the upper and lower teeth naturally together, the tongue tip against the premaxillary teeth, and with all patients breathing quietly

without swallowing or chewing. Philips DICOM viewer software was used to measure CT parameters including the angles, cross-sectional areas, and soft tissue parameters of the upper airway: the sella (S), nasion (N), subspinale (A), supremental (B), basion (Ba), vallecula (V), tip of tongue (T), anterior nasal spine (ANS), posterior nasal spine (PNS), angles relative to each other (e.g., SNA, SNB, BaSN, and ANB), and the posterior airway space (Fig. 2). Then, the sagittal (SD) and coronal (CD) diameters and cross-sectional area were measured at the palatopharynx, glossopharynx , hypopharynx, and the narrowest part of the glossopharynx (*Sakat et al., 2016*) (Fig. 3). In addition, the soft tissues of the upper airway were measured, including the vallecula-tip of tongue (VT, the tongue length), soft palate length, and soft palate thickness (SPT).

### Statistical analysis

The statistical program SPSS 20.0 (IBM, Armonk, NY, USA) was used to compare the data associated with anthropometric characteristics, upper airway morphology, and PSG between the POSA and NPOSA groups. Normally distributed data were compared using the independent samples $t$-test and are presented as the mean ± standard deviation. Non-normally distributed data were analyzed using the Mann–Whitney test and are presented as the median (interquartile range). Correlations between variables were examined using Spearman's correlation or the Pearson test. Forward logistic regression analysis was performed to select the main correlative parameters of POSA. The accuracy of the diagnostic model was examined using the receiver operating characteristics (ROC) curve. A value of $p < 0.05$ was used to indicate statistical significance.

## RESULTS

The 75 subjects were divided into NPOSA ($n = 40$) and POSA ($n = 35$) groups. The mean age of the subjects was 39.2 ± 9.4 years, while the mean age of the NPOSA and POSA groups was 40.8 ± 5.3 and 38.6 ± 11.2 years ($p = 0.043$). The mean BMI was 27.4 ± 3.2 kg/m$^2$. There was a significant ($p < 0.01$) difference in the AHI between NPOSA and POSA (60.1 ± 19.4 *vs.* 42.5 ± 18.5 event/h), and the L-AHI of POSA was significantly ($p < 0.001$) lower than that of NPOSA. However, no significant difference between the two groups was found for BMI, tongue position, tonsil size, Friedman staging, or S-AHI ($p > 0.05$) (Table 1).

The SPL was significantly ($p = 0.005$) longer in NPOSA (36.1 ± 5.0 mm) than in POSA (33.0 ± 3.9 mm). The cross-sectional area of the palatopharynx was significantly ($p = 0.027$) smaller in the NPOSA group (67.4 (31.1) *vs.* 80.3 (43.0) mm$^2$). In addition, the CD of the palatopharynx and the narrowest part of the glossopharynx were significantly ($p < 0.05$) smaller in NPOSA than in POSA. However, there were no statistical differences in SNA, SNB, ANB, BaSN, ANS, PNS, SPT, VT, or the SD to CD ratios of the glossopharynx and hypopharynx (all $p > 0.05$) between the two groups. Although, no significant difference group was found in CD or the cross-sectional area of the glossopharynx or hypopharynx, the values were smaller in NPOSA than POSA (Table 2).

After adjusting for age as a confounding factor, the AHI, SPL, and SD to CD ratios of the glossopharynx and L-AHI were significantly ($p < 0.05$) larger in NPOSA than POSA, and

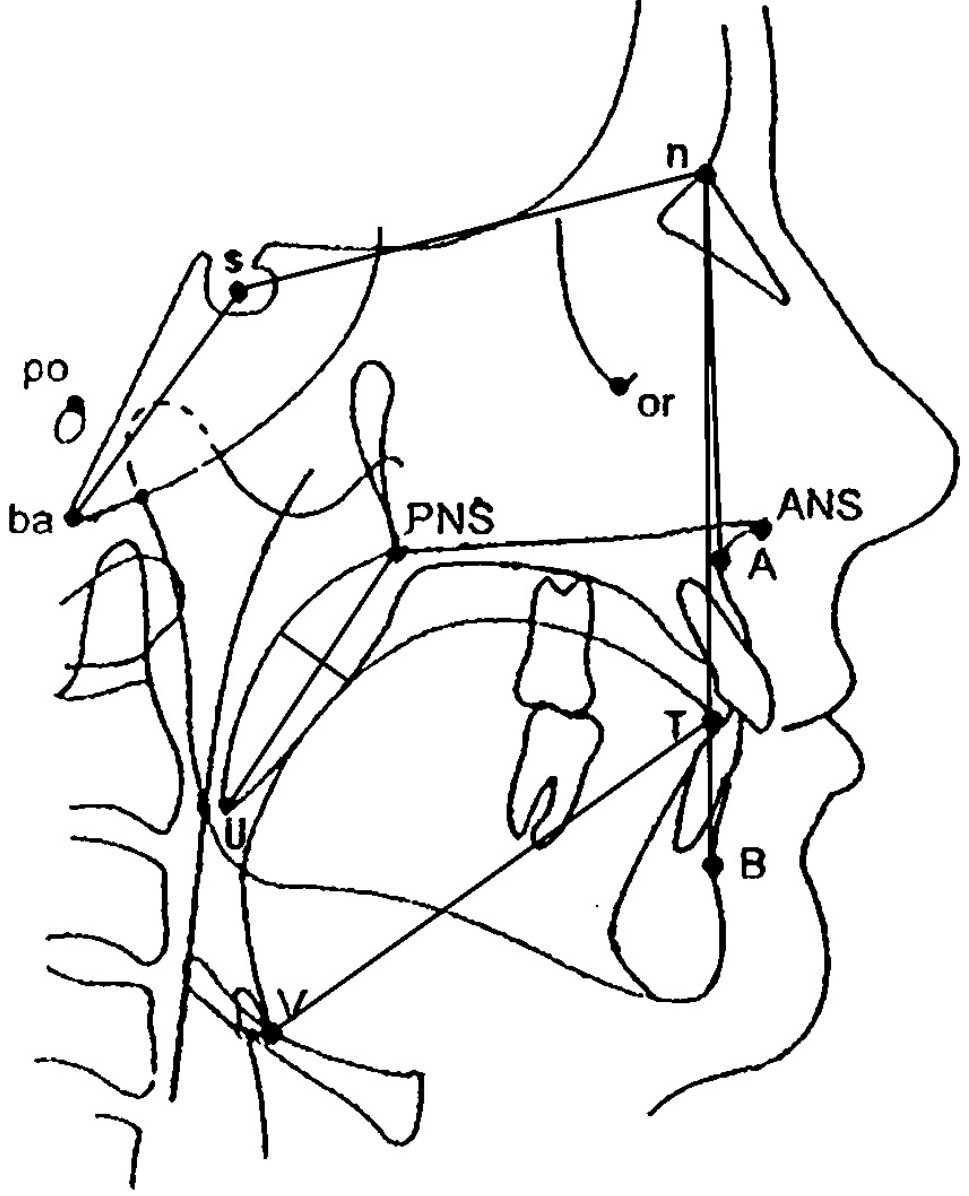

**Figure 2  Diagram showing the anatomical points, lines, and angles used to evaluate craniofacial morphology.** s, sella; n, nasion; A, subspinale; B, supremental; ba, basion; V, vallecula; T, tip of tongue; ANS, anterior nasal spine; PNS, posterior nasal spine.

the CD of the glossopharynx was significantly smaller. However, no significant difference was found in CD or the cross-sectional area of the palatopharynx between the two groups.

As assessed using Pearson and Spearman correlation analyses, L-AHI was correlated with the cross-sectional area ($r = -0.306$, $p = 0.008$) and CD ($r = -0.398$, $p < 0.001$) of the palatopharynx, the cross-sectional area ($r = -0.241$, $p = 0.038$) and CD ($r = -0.297$, $p = 0.010$) of the narrowest part of the glossopharynx, the CD of the glossopharynx
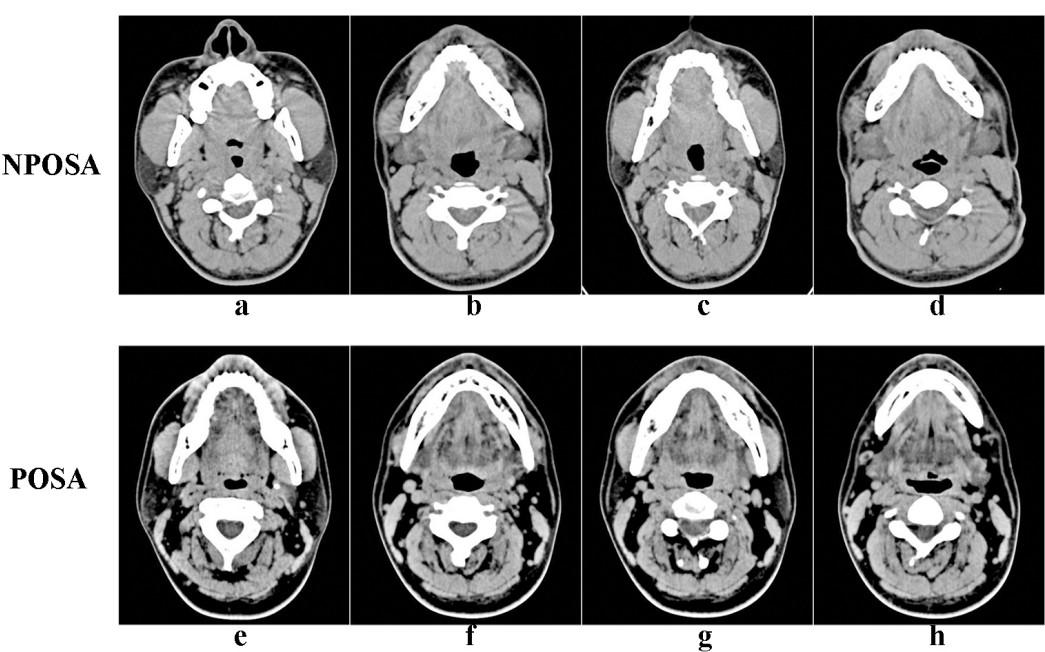

**Figure 3 Cross-sectional views of the upper airway in the four planes in NPOSA (A–D) and POSA (E–H).** NPOSA, non-positional obstructive sleep apnea; POSA, positional obstructive sleep apnea; (A), (E), cross-sectional views of the palatopharynx; (B), (F), cross-sectional views of the glossopharynx; (C), (G), cross-sectional views of the hypopharynx; (D), (H), the cross-sectional views of the narrowest part of the glossopharynx.

($r = -0.284$, $p = 0.013$), BMI ($r = 0.273$, $p = 0.018$), SPL ($r = 0.284$, $p = 0.014$), and VT ($r = 0.250$, $p = 0.030$) in all subjects.

Forward logistic regression analysis of the relative influence of these independent variables on NPOSA, the soft palate length, the CD of the narrowest part of the glossopharynx, and age were included in the regression model (Table S1). The cut-off point for soft palate length was 35.4 mm.

## DISCUSSION

The AHI is significantly higher in NPOSA than POSA (*Oksenberg et al., 1997*), consistent with our results. Aging is known to be a major contributor to the risk of OSAHS; its prevalence and severity increase with age and may be associated with factors such as a reduction in muscle tone, a reduced airway lumen, and changes in lung volume and ventilatory control stability (*Edwards et al., 2010*; *Kim et al., 2015*). In the present study, the NPOSA patients were older than the POSA patients. Although there was no correlation between age and AHI or L-AHI, the difference in upper airway parameters between the two groups changed after adjusting for age. Age was included in the regression model as one of the independent risk factors for NPOSA. This may be because the activity of the pharyngeal dilator muscles in response to negative pharyngeal pressure is impaired upon aging, increasing the compliance of the pharyngeal wall in both the supine and lateral
**Table 1  Comparison of the data associated with anthropometric and cephalometric characteristics and polysomnography (PSG) in positional (POSA) and non-positional (NPOSA) obstructive sleep apnea.**

| Indicators | [c]NPOSA (45) | [c]POSA (30) | $p$[a] | $p$[b] |
|---|---|---|---|---|
| Age (year) | 40.1 ± 9.0 | 36.6 ± 9.6 | 0.043 | – |
| BMI (kg/m$^2$) | 27.9 ± 3.1 | 26.6 ± 3.1 | 0.106 | 0.069 |
| SNA (°) | 83.4 ± 4.2 | 82.3 ± 3.8 | 0.254 | 0.148 |
| SNB (°) | 79.1 ± 4.5 | 78.5 ± 4.6 | 0.539 | 0.289 |
| ANB (°) | 4.4 (2.5) | 3.9 ± 2.7 | 0.479 | 0.666 |
| BaSN (°) | 139.1 ± 5.3 | 140.6 ± 5.3 | 0.222 | 0.134 |
| PAS (mm) | 11.3 (3.9) | 10.8 ± 3.2 | 0. 449 | 0.288 |
| ANSPNS (mm) | 45.2 ± 3.8 | 44.8 ± 2.9 | 0.617 | 0.576 |
| SPL (mm) | 36.1 ± 4.9 | 33.0 ± 3.9 | 0.005 | 0.005 |
| SPT (mm) | 10.2 ± 2.0 | 9.6 ± 1.5 | 0.159 | 0.219 |
| VT (mm) | 71.2 ± 5.8 | 69.1 ± 4.9 | 0.106 | 0.094 |
| Total AHI (events/h) | 60.1 ± 19.4 | 42.5 ± 18.5 | <0.001 | <0.001 |
| S-AHI | 62.0 ± 20.3 | 64.0 (20.1) | 0.405 | 0.347 |
| L-AHI | 60.1 ± 23.5 | 9.5 (11.6) | <0.001 | <0.001 |

**Notes.**

[a]Mann–Whitney test or $t$ test for equal change between groups.

[b]Test for equal change between groups, adjusted for age.

[c]Normally distributed data are presented as mean ± standard deviation; non-normally distributed data are presented as median (interquartile range).

NPOSA, non-positional obstructive sleep apnea; POSA, positional obstructive sleep apnea; BMI, body mass index; SPL, soft palate length; SPT, soft palate thickness; VT, the tongue length; AHI, apnea hypopnea index; S-AHI, supine-AHI; L-AHI, lateral-AHI.

positions. Furthermore, age may affect upper airway collapse in different planes and play a role in the susceptibility to NPOSA.

Cephalometry is used to screen for anatomical abnormalities in OSAHS patients. Various studies have shown that the pharyngeal dimensions are correlated with both craniofacial skeletal morphology (*Bacon et al., 1988*; *Wang et al., 2014*; *Yu, Fujimoto & Urushibata, 2003*) and airway obstruction in OSAHS patients (*Ardehali et al., 2016*), including those with ANB (*Ceylan & Oktay, 1995*), SNB (*Ardehali et al., 2016*), and ANS-PNS (*Yu, Fujimoto & Urushibata, 2003*). It is well-known that significant differences are apparent between NPOSA and POSA in terms of clinical symptoms, and, theoretically, the characteristics of the craniofacial skeleton may play a role in this context. However, we found no significant difference in craniofacial morphology (apart from soft palate length) between NPOSA and POSA, suggesting that skeletal morphology did not play a significant role in NPOSA.

We found that the L-AHI was associated with soft palate length; specifically, the more the length of the soft palate exceeded > 35.4 mm the greater the risk of NPOSA. This may be explained in several ways. Historically, the soft palate was larger in OSAHS than non-OSAHS patients (*Lowe et al., 1996*), and has been shown to play a key role in OSAHS (*Bacon et al., 1990*). In addition, fatty infiltration (regardless of obesity status) (*Zohar et al., 1998*) impairs soft palate sensation, which correlates with the severity of snoring (*Jeong et al., 2016*); the soft palate becomes altered and exhibits a fiber-type appearance caused by the additional load on the velopharyngeal muscles (*Lindman & Stal, 2002*) in OSAHS

**Table 2** Comparison of the data associated with the cephalometric characteristics in POSA and NPOSA.

| | Variables | [c]NPOSA(45) | [c]POSA(30) | $p$[a] | $p$[b] |
|---|---|---|---|---|---|
| Palatopharyngeal | Cross-sectional area | 67.4 (31.1) | 80.3 (43.0) | 0.027 | 0.107 |
| | SD | 8.1 (3.5) | 8.5 ± 2.0 | 0.563 | 0.940 |
| | CD | 8.8 (3.8) | 9.7 (5.2) | 0.040 | 0.124 |
| | SD/CD | 0.9 (0.6) | 0.9 ± 0.3 | 0.320 | 0.666 |
| Glossopharynx | Cross-sectional area | 287.4 ± 86.6 | 311.5 ± 94.0 | 0.257 | 0.363 |
| | SD | 16.3 ± 3.8 | 15.9 (6.6) | 0.888 | 0.955 |
| | CD | 21.8 ± 6.4 | 24.4 ± 5.5 | 0.076 | 0.103 |
| | SD/CD | 0.7 (0.4) | 0.6 (0.2) | 0.261 | 0.242 |
| Hypopharynx | Cross-sectional area | 123.8 (55.6) | 153.2 ± 61.5 | 0.098 | 0.441 |
| | SD | 10.7 (2.5) | 10.9 ± 2.7 | 0.516 | 0.793 |
| | CD | 15.7 ± 5.9 | 17.2 ± 5.3 | 0.276 | 0.218 |
| | SD/CD | 0.7 (0.5) | 0.7 ± 0.3 | 0.706 | 0.155 |
| The narrowest level of glossopharynx | Cross-sectional area | 189.0 ± 82.3 | 211.5 ± 69.1 | 0.222 | 0.177 |
| | SD | 16.1 ± 5.1 | 14.9 (6.9) | 0.393 | 0.347 |
| | CD | 11.5 (9.3) | 14.5 ± 6.0 | 0.055 | 0.029 |
| | SD/CD | 1.2 (1.6) | 1.1 (0.9) | 0.119 | 0.042 |

Notes.
[a]Mann–Whitney test or $t$ test for equal change between groups.
[b]Test for equal change between groups, adjusted for age.
[c]Normally distributed data are presented as mean ± standard deviation; non-normally distributed data are presented as median (interquartile range).
NPOSA, non-positional obstructive sleep apnea; POSA, positional obstructive sleep apnea; SD, sagittal diameter; CD, coronal diameter; SD/CD, sagittal diameter to coronal diameter ratios.

patients. These factors may affect compliance of the soft palate and can cause narrowing of the upper airway. Also, inflammatory processes increase the thickness of the soft palate (*Berger et al., 2002*), which also tends to be thicker in the supine position in OSAHS patients because gravitational force reduces the SD of the upper airway (*Lowe et al., 1996*). In comparison, in the lateral position, the gravitational force acting on the longer soft palate of NPOSA patients should decrease the CD of the upper airway. Furthermore, our simplified PRE screening model included soft palate length as an independent predictor of NPOSA. Our results suggest that soft palate length plays a significant role in the direction of upper airway collapse during sleep, although the reason for the increased susceptibility to NPOSA in those with longer soft palates remains unclear. Our work suggests that NPOSA not only involves anteroposterior collapse, but also transverse collapse in the plane of the palatopharynx, whereas POSA involves only the former.

In OSAHS patients, single or multiple planes of upper airway collapse, such as the palatopharyngeal, glossopharyngeal, and hypopharyngeal planes (*Tang et al., 2012*; *Torre et al., 2016*; *Vroegop et al., 2014*), and the complete concentric collapse of the glossopharynx, are correlated significantly with the severity of OSAHS (*Ravesloot & De Vries, 2011*; *Schwartz et al., 2015*; *Vroegop et al., 2014*). In our study, the negative correlations between L-AHI and the CD of the palatopharynx and glossopharynx demonstrated that the lateral diameter of the upper airway was highly predictive of the value of AHI (*Tsai et al., 2003*).

Second, after adjusting for age, patients with NPOSA tended to have more severe OSAHS, with a smaller CD at the narrowest level of the glossopharynx and a higher L-AHI. Third, the CD of the narrowest level of the glossopharynx, age, and SPL were included in the regression model. These results indicate that the smaller lateral distance in NPOSA patients (*Pevernagie et al., 1995*; *Soga et al., 2009*), particularly at the level of the glossopharynx, plays an essential role in the differences between NPOSA and POSA. There was no significant difference in the SD of the upper airway or S-AHI between the two groups, suggesting that the pathogenesis of NPOSA in the supine position might be similar to that of POSA. Additionally, in NPOSA, a round airway was seen in the palatopharynx and an elliptical airway, with the SD as the long axis, was seen at the narrowest level of the glossopharynx. In the supine position in POSA, an elliptical airway with the CD as the long axis was seen in the palatopharynx (*Ciscar et al., 2001*; *Walsh et al., 2008*). Perhaps the different airway shapes between NPOSA and POSA, caused by the fat distribution in the lateral pharyngeal wall and the reduction in muscle tone (*Saigusa et al., 2009*), led to differences in the direction of upper airway collapse and impaired breathing in sleep (*Foster et al., 2009*). Hence, the upper airway morphology characteristics of the two groups may play a significant role in the substantial differences between NPOSA and POSA.

Our small sample size and the lack of a control group are limitations of our study. Also, the patients were very young and the results might thus not be generalizable. Moreover, we did not collect nocturnal rostral fluid shift or electroencephalographic data, and may also have ignored other factors that could affect our results. Such limitations are disadvantages common to many retrospective studies. Another limiting factor is that CT was performed with all patients awake and the data may thus not accurately reflect the dynamics of upper airway collapsibility during sleep. Thus, future studies with larger sample numbers are warranted to further examine and prospectively elucidate the mechanisms underlying NPOSA and POSA.

This study identified characteristic differences in the upper airway between NPOSA and POSA, which may partly explain the differences in clinical characteristics and treatment success rates between the two groups. The prognosis of OSAHS may be improved by choosing the appropriate treatment according to the upper airway morphology characteristics. POSA patients may gain more benefit from positional therapy (*De Vries et al., 2015*; *Levendowski et al., 2014*), mandibular advancement devices, (*Lee et al., 2012*) and uvulopalatopharyngoplasty (*Li et al., 2013*), whereas NPOSA patients will probably almost always require continuous positive airway pressure therapy and/or more complex treatment.

## CONCLUSION

In terms of upper airway morphology, the NPOSA group had a smaller pharyngeal CD and a longer soft palate than the POSA group. In addition to the AHI, the main differences between NPOSA and POSA patients were in the soft palate length and the CD at the narrowest part of the glossopharynx.

## ACKNOWLEDGEMENTS

The authors would like to thank all subjects who participated in the study.

### Funding

This work was supported by grants-in-aid from the Shanghai Science and Technology Commission Project of Shanghai (124119a9700), the Shanghai Shen-Kang Hospital Management Center Project of Shanghai (SHDC12015101) and the National Natural Science Foundation of China (81500780). The funders had no role in study design, data collection and analysis, decision to publish, or preparation of the manuscript.

### Grant Disclosures

The following grant information was disclosed by the authors:
Shanghai Science and Technology Commission Project of Shanghai: 124119a9700.
Shanghai Shen-Kang Hospital Management Center Project of Shanghai: SHDC12015101.
National Natural Science Foundation of China: 81500780.

### Competing Interests

The authors declare there are no competing interests.

### Author Contributions

- Xiao Jiao conceived and designed the experiments, performed the experiments, analyzed the data, contributed reagents/materials/analysis tools, wrote the paper, prepared figures and/or tables.
- Jianyin Zou performed the experiments, analyzed the data.
- Suru Liu and Jian Guan contributed reagents/materials/analysis tools.
- Hongliang Yi conceived and designed the experiments, analyzed the data, contributed reagents/materials/analysis tools, reviewed drafts of the paper.
- Shankai Yin contributed reagents/materials/analysis tools, reviewed drafts of the paper.

### Human Ethics

The following information was supplied relating to ethical approvals (i.e., approving body and any reference numbers):

This project was approved by the Ethics Committee of Shanghai Jiao Tong University Affiliated Sixth People's Hospital and complied with the Declaration of Helsinki.

### Data Availability

The raw data is included as Data S1.

### Supplemental Information

Supplemental information for this article can be found online at http://dx.doi.org/10.7717/peerj.3918#supplemental-information.

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
