# Peer review of "A retrospective study: does upper airway morphology differ between non-positional and positional obstructive sleep apnea?"

_PeerJ, doi:10.7717/peerj.3918_

## Round 0.1 · original submission · Major Revisions

· Academic Editor

Major Revisions

Dear Authors,

The Reviewers found your manuscript very interesting but raised several concerns that I would suggest you to take into consideration and discuss or incorporate into your manuscript in order to reach the level requested for publication.

Best regards
Salvatore Andrea Mastrolia

·

Basic reporting

I think the article can be published with corrections. Given the limitations of a retrospective study, the reported data and the conclusions still make sense. The authors used routine polysomnography and CT scanning in the appropriate manner and the data were unsurprising. The fact that NPOSA , compared to POSA, is worse and due to anatomical abnormalities is not at all a surprise. And the reasons appear totally logical in that there was, on the average, a lengthened soft palate and a reduced or narrower glossopharynx.

My concern is with the clinical significance with respect to treatment regarding these observations. If one has OSA it requires treatment regardless of whether one has POSA or NPOSA due to the well-known debilitating consequences of untreated OSA. One possible conclusion is that NPOSA would probably almost always require PAP therapy whereas POSA might be more susceptible to a mandibular repositioning device. Some discussion along these lines might be helpful.

Wrt data reporting, there is no need for 2 decimal points. For example, patient age should be 30.2 +/- 9.4 years not 39.23 +/- 9.41 years. And the same applies to BMI, AHI, and so on.

Experimental design

They authors used routine Procedures to assess OSA and to measure the patients' anatomical characteristics. And the statistical evaluation followed standard and correct procedures.

Validity of the findings

The findings are valid while not being very surprising. However, nice to have meaningful statistical data.

Additional comments

The comments above should suffice but happy to make further comment if the review is unclear in any way.

Reviewer 2 ·

Basic reporting

The background needs to demonstrate the importance of this work.

Experimental design

The method of research needs to be detailed.

Validity of the findings

no comment

Additional comments

This study is to investigate the differences in upper airway morphology between positional (POSA) and non-positional (NPOSA) in 75 patients with obstructive sleep apnea. But please consider the following questions:

1.What is the significance of exploring this difference in the progress and treatment of the patients with OSA?

2.How to judge POSA and NPOSA? You should provide for the method of analysis of these concepts.

3.How to consider the relationship between nocturnal rostral fluid shift and the severity of OSA? How to avoid this kind of confounding factors?

4.Why all of these patients only have OSA? How about MSA or CSA these patients?

·

Basic reporting

Jiao and colleagues retrospectively described the differences in upper airway morphology between 30 patients with positional and 45 with non-positional OSA. Various measurements and diameters calculated on CT scan are compared between the 2 groups.
The paper is overall well written and shows some significant results, however there are some major and minor revisions to address. In particular, the authors use too many abbreviations, and some of them are not so easily interpreted, which makes the article sometimes difficult to read, since the reader is obliged to check the meaning of the abbreviation quite often. Furthermore, The discussion is poorly structured, I suggest to briefly report the main results and then discuss each of them comparing with prior literature.

Major revisions:
- Materials and Methods: the authors state that they performed a PSG, but they did not include electroencephalographic data. I suggest to add this among the study limitations.
- Results: Mean age of study population was 39 years. Differently from other studies, The study population is very young, can the authors justify and comment on this?
- in the Results, line 119, the authors do not write if BMI was different between groups. This data is written in table 1, however, given its importance, I suggest to add it in the text, too.
- discussion, lines 154-157: the meaning of this sentence is not clear, I suggest rephrasing.
- discussion, Lines 159-166: I suggest to deepen this point.
- discussion, line 178-179: 'specifically...35.41mm'. This result is not present in the result section. I suggest to add it to the section and then discuss.
- discussion, line 196-199: ' while there... the two groups': I suggest to better explain this point.
- discussion: add to the limits of the study the retrospective nature.

Experimental design

Major revisions:
- CT test is not routinely asked in patients with OSA. The authors should explain why the exam was requested in this group of patients. If it was requested for research purposes, patients should have signed an informed consent.

Validity of the findings

no comment

Additional comments

Minor revisions:
- introduction line 59: few studies on the Chinese population. Line 60: Chinese patients using...
- Materials and Methods line 68: treated in the hospital
- Materials and Methods line 93: would have had calming
- Results line 133: GP was significantly smaller.
- conclusions, line 217: were in SPL, and the CD...
- figure 1: the position of the head during CT was not in the correct position
- figure 1: >25% of all.

---

## Round 0.2 · Major Revisions

· Academic Editor

Major Revisions

Dear Authors,

The Reviewer appreciated and recognized your efforts in improving your manuscript, but again underlined that there is a major limitation due to the written English that is still not reaching the standards for publication.

I would kindly recommend you to refer to a proof reading service in order to improve the readability of your manuscript and achieve publication.

Best regards,
Salvatore Andrea Mastrolia
PeerJ Academic Editor

·

Basic reporting

I acknowledge that the authors have made some corrections and improvements to the original manuscript, however most of the statements that were not clear in the first version are still not clear. The manuscript absolutely needs to be revised by an English translator, in fact, many statements sound incomplete and without the main verb.

Minor comment:
- Materials and Methods: the authors state that the 105 patients underwent surgery. I suggest to add the reason why they required surgery (OSAS treatment)

Experimental design

no further comments

Validity of the findings

no further comments

---

## Round 0.3 · accepted · Accept

· Academic Editor

Accept

Dear Authors,

I would like to compliment with you for the efforts provided in addressing the Reviewers' comments.

I fell that your manuscript has reached the level of publication and can be accepted in its current form.

Best regards

Salvatore Andrea Mastrolia
PeerJ Academic Editor